# First known trace fossil of a nesting iguana (Pleistocene), The Bahamas

**Anthony J. Martin**[1]*, **Dorothy Stearns**[2], **Meredith J. Whitten**[3], **Melissa M. Hage**[4], **Michael Page**[1,5], **Arya Basu**[5]

**1** Department of Environmental Sciences, Emory University, Atlanta, Georgia, United States of America, **2** School of Medicine, University of Colorado, Aurora, Colorado, United States of America, **3** North Carolina Division of Marine Fisheries, Morehead City, North Carolina, United States of America, **4** Department of Environmental Sciences, Oxford College of Emory University, Oxford, Georgia, United States of America, **5** Center for Digital Scholarship, Emory University, Atlanta, Georgia, United States of America

* geoam@emory.edu

**Data Availability Statement:** All relevant data are within the manuscript and its Supporting Information files.

**Funding:** The authors received no specific funding for this work.

## Abstract

Most species of modern iguanas (Iguania, Iguanidae) dig burrows for dwelling and nesting, yet neither type of burrow has been interpreted as trace fossils in the geologic record. Here we describe and diagnose the first known fossil example of an iguana nesting burrow, preserved in the Grotto Beach Formation (Early Late Pleistocene, ~115 *kya*) on San Salvador Island, The Bahamas. The trace fossil, located directly below a protosol, is exposed in a vertical section of a cross-bedded oolitic eolianite. Abundant root traces, a probable land-crab burrow, and lack of ghost-crab burrows further indicate a vegetated inland dune as the paleoenvironmental setting. The trace fossil matches dimensions and overall forms of burrows made by modern iguanas, and internal structures indicate active backfilling consistent with modern iguana nesting burrows. The trace fossil is also located on an island with a modern native species of rock iguana (*Cyclura riyeli riyeli*), suggesting a presence of iguanas on San Salvador since the Late Pleistocene. This nesting burrow may provide a search image for more fossil iguana burrows in The Bahamas and other places with long-established iguana species and favorable geological conditions for preserving their burrows.

## Introduction

Iguanas (Iguania, Iguanidae) are relatively large and mostly herbivorous and terrestrial lizards that are primarily native to Central America, South America, the Caribbean, and The Bahamas [1–4]. Iguanian body fossils support a phylogenetic history dating back to at least the Cretaceous Period, and their skeletal record is well represented in Paleogene, Neogene, and Quaternary rocks [5–16]. However, trace fossils attributed to iguanas, whether as tracks, burrows, coprolites, or additional sign, remain unreported. We assume this apparent paucity of iguana trace fossils is more a result of non-recognition rather than actual scarcity, particularly because modern iguanas construct both dwelling and nesting burrows, which may outnumber iguanas living in a given area [3, 17–24]. Burrowing also bestows reproductive and survival advantages for most modern iguana species, as burrows enhance fitness, juvenile and adult longevity, and

**Competing interests:** The authors have declared that no competing interest exist.

population numbers [3, 17, 18, 20, 22, 25, 26]. The evolutionary history of iguanas accordingly should reflect the products of burrowing behaviors as trace fossils preserved in terrestrial sediments and within their paleogeographic ranges. Furthermore, owing to the abundance, depths, and lengths of iguana burrows [17–22], these structures should have excellent preservation potential as trace fossils.

Thus we are pleased to report the first known trace fossil of an iguana nesting burrow, preserved in a Middle-Late Pleistocene (*circa* 115 *kya*) oolitic eolianite on San Salvador Island (The Bahamas). The trace fossil is comparable in size and form to modern iguana nesting burrows, has internal structures consistent with nesting behavior, and is preserved on the margin of a former eolian dune, which is an appropriate facies setting for iguana nesting. Moreover, a terrestrial and burrow-nesting rock iguana (*Cyclura riyeli riyeli*) lives on San Salvador and surrounding cays [22, 27–29]. Based on Late Pleistocene-Holocene body fossils of *C. riyeli riyeli* on San Salvador [7, 8], these modern populations were preceded by ancestral ones, further supporting the likelihood of iguana trace fossils in the geologic record there. If confirmed, this trace fossil would also establish a minimum presence of iguanas on San Salvador since the Middle-Late Pleistocene, which is paleobiogeographically significant, as San Salvador is isolated from the main Bahamas Platform [30–32]. Our discovery further encourages hope that more such burrows are preserved in other Pleistocene eolianites on San Salvador and elsewhere in The Bahamas.

## Materials and methods

The trace fossil was first noticed by one of us (AJM) in a road cut on the southern end of San Salvador Island (The Bahamas) on December 29, 2013, which was witnessed by two of the coauthors (MJW and DS). Two of us (AJM and MJW) initially documented the structure on January 5, 2014 through preliminary measurements and photographs. On January 3, 2016, AJM and another coauthor (DS) revisited the site and re-measured, described, and photographed the structure more extensively. A third visit by AJM with another coauthor (MMH) on March 15, 2018 resulted in more insights into the sedimentological setting of the structure, as well as the discovery of a significant invertebrate trace fossil nearby. This last visit is also when the structure was photographed for photogrammetry.

Field methods involved use of a grain-size chart and hand lens to describe the gross lithology of the outcrop, and a tape measure to define outcrop width and height, as well as dimensions of the investigated structure. Field descriptions and labeled sketches were made onsite and later augmented by labeled drawings based on photographs. During our third visit to the outcrop, on March 15, 2018, one of us (AJM) in the presence of another (MHH) took 74 overlapping high-resolution photographs of the specimen with a handheld Nikon CoolPix A900 digital camera, which were used later for photogrammetric processing. For this, we first fixed seven unique paper photogrammetric markers onto the outcrop with duct tape (S1 Fig). Four markers were placed vertically at 15-cm intervals about one meter left of the specimen, two were placed vertically one meter right of the specimen and at the same level as the middle two left, and one was put directly below the specimen. Photographs were taken from the same spot by moving the camera from left to right, then right to left, panning from the top portion to bottom portion for maximum overlap. AgiSoft Photoscan™ (now called AgiSoft Metashape™) photogrammetry software was later used to stitch the photographs together and render a three-dimensional rotatable image that better represents the overall form of the structure (S2–S4 Figs in S1 File).

The January 2016 visit by two of us (AJM and DS) was noteworthy for a boulder field 100–150 m west of the outcrop that represented a major alteration of local environments. These boulders were derived from coastal outcrops south of the road cut and transported by

Hurricane Joaquin, which hit San Salvador Island on October 2, 2015 [33, 34]. Fortunately, the road cut and structure examined in this study seemed relatively unaffected by the storm, an assessment later verified by our comparing field photographs of the outcrop and structure in 2014, 2016, and 2018.

## Results

### Study location and stratigraphy

The study location is near the southeastern end of San Salvador Island, The Bahamas in an area called The Gulf, at N 23˚ 56.832', W 74˚ 30.654' (Fig 1). The specimen is exposed in a vertical section of an east-west trending road cut about 2 m north of Queen's Highway and about 35 m north of a high-energy rocky shoreline. Based on previous geologic mapping of this area, the probable stratigraphic position of the outcrop is in regressive eolianite facies toward the top of the Cockburn Town Member, which is the uppermost member of the Grotto Beach Formation (Fig 2) [30, 31, 35, 36].

The outcrop hosting the structure is above a paleosol in the Cockburn Town Member, but stratigraphically below a prominent *terra rosa* paleosol exposed along the rocky coastline south of the outcrop. Radiometric dating has not been applied to this or nearby outcrops in The Gulf, and index fossils do not provide precise assessments of their age, either. However, its stratigraphic context with relation to other strata on San Salvador–particularly paleosols below and above it–indicate a probable age from the earliest part of the Late Pleistocene (Tarantian) at about 115 *kya* [30, 31, 36–39]. Facies are interpreted as part of a regressive eolian sequence following the 5e sea-level high (~125-*kya*) recorded throughout The Bahamas [30, 31, 36–39]. Exposed strata in the area superficially resemble younger eolianite facies of the Holocene Rice Bay Formation, which crop out extensively in the northeastern corner of San Salvador. Those deposits, though, were formed as transgressive coastal dunes only 3–5 *kya* [30, 31, 36, 39].

### Lithology and physical sedimentary structures

The outcrop hosting the structure has a broad, convex, lens-like profile exposed in a 70–75 m straight-line distance parallel to the road. The outcrop is 2.9 m high in its middle, with a weathered layer of rubble on top (Fig 3A); it tapers at road level on both western and eastern ends as well as to its north, resulting in an isolated partial dome. This portion of the road cut was also formerly connected to a less lengthy (50–55 m long) and shorter (< 2 m tall) outcrop about 10 m south, having been separated by the original construction of Queen's Highway. The structure investigated in this study is 10.5 m west of the easternmost edge of the northern outcrop (Fig 3B).

The main lithology is a cross-bedded oolitic grainstone (limestone) with minor amounts of bioclasts and lithoclasts. Grain sizes are dominantly very fine to medium sand, but with pebble-sized lithoclasts in some beds. The most prominent physical sedimentary structures in the lithology are low- to high-angled planar cross-beds, with apparent dips as much as 20˚ where exposed (Fig 3B).

### Biogenic sedimentary structures

Bedding is disrupted in places by small-diameter (5–10 mm) branching structures oriented vertically to obliquely, and tapering downward and distally (Fig 4A and 4B). These structures, preserved in semi- and full-relief, resemble modern plant roots. They are identified accordingly as root trace fossils, which are also common in eolianites of the Cockburn Town Member in the same area [40–44]. Root trace fossils are most abundant in a 50-cm thick area immediately beneath a thin (5–10 mm) white micritic layer in the thickest portion of the outcrop.

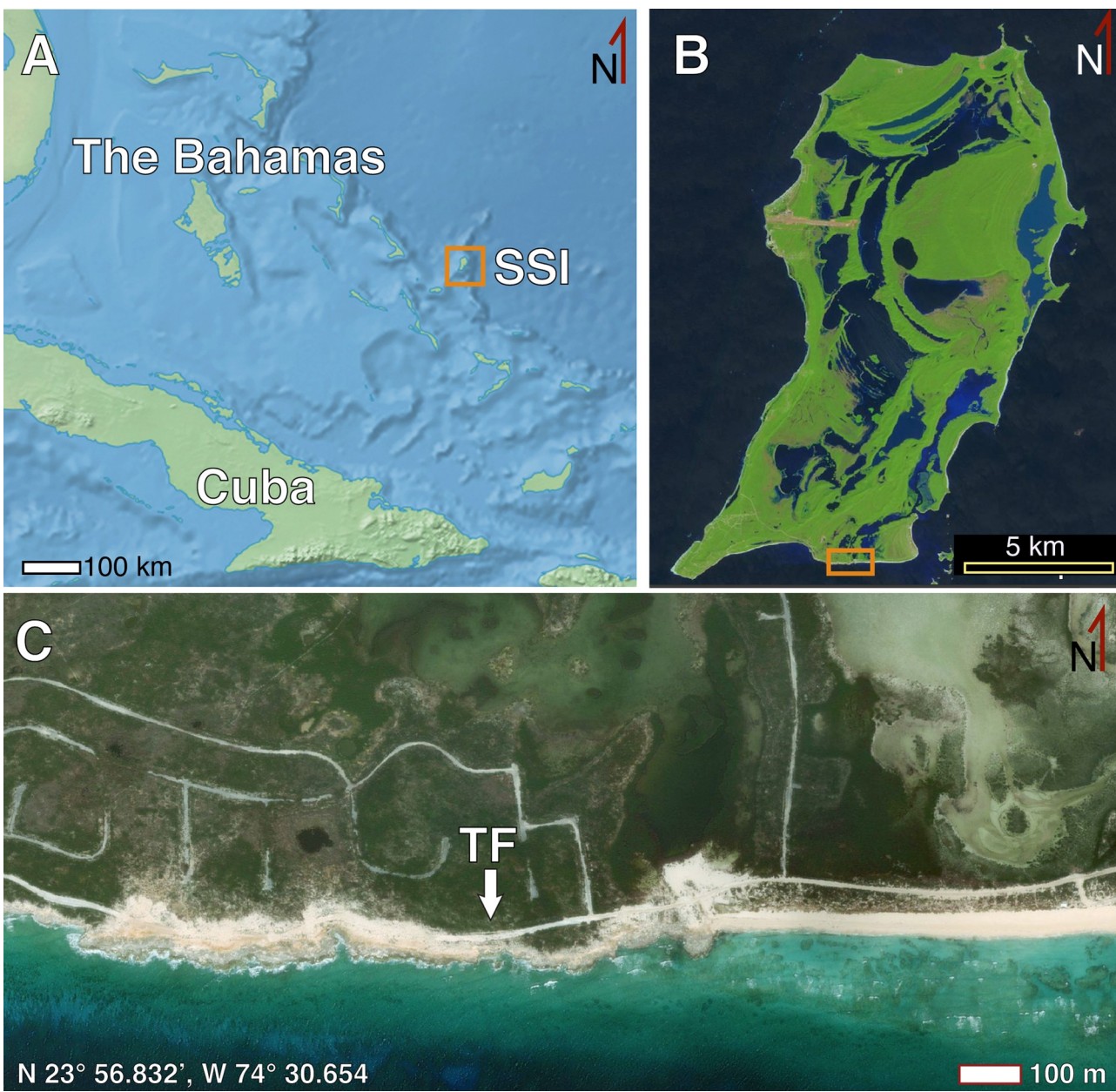

**Fig 1. Location of trace fossil in this study.** (A) The Bahamas archipelago north of Cuba and east of Florida, with San Salvador Island (SSI) indicated. (B) San Salvador Island, with study area on southern end of island (box); dark- and light-blue areas are inland lakes. (C) Location of trace fossil (TF) investigated in this study in roadcut north of The Gulf, at N 23° 56.832', W 74° 30.654.' Image in Fig 1A downloaded from Natural Earth (https://www. naturalearthdata.com/), released into public domain and retrieved October 29, 2020, used in Photoshop. Image in Fig 1B from European Space Agency (ESA) Sentinel-2 Satellite, January 2019 image of San Salvador Island from IDL1C_T18QWM_A018555_20190110T154545, (https://earthexplorer.usgs. gov/), released in public domain and retrieved October 30, 2020, used in QGIS. Image in Fig 1C from Microsoft Bing Image Tile of San Salvador Island, Bahamas (http://ecn.t3.tiles.virtualearth.net/tiles/a{q}.jpeg?g=1), downloaded through QGIS with image service provided by Microsoft (open source), retrieved October 31, 2020 and used in QGIS. No logos or trademarks were removed; annotations are by the authors.

Despite its thinness, this micritic layer is nearly continuous throughout most of the outcrop, following the convex contour of the top surface.

A few thin (2–5 mm wide) and short (3–7 cm long) vertically oriented tube-like structures, interpreted as passively filled fossil invertebrate burrows, cut across bedding

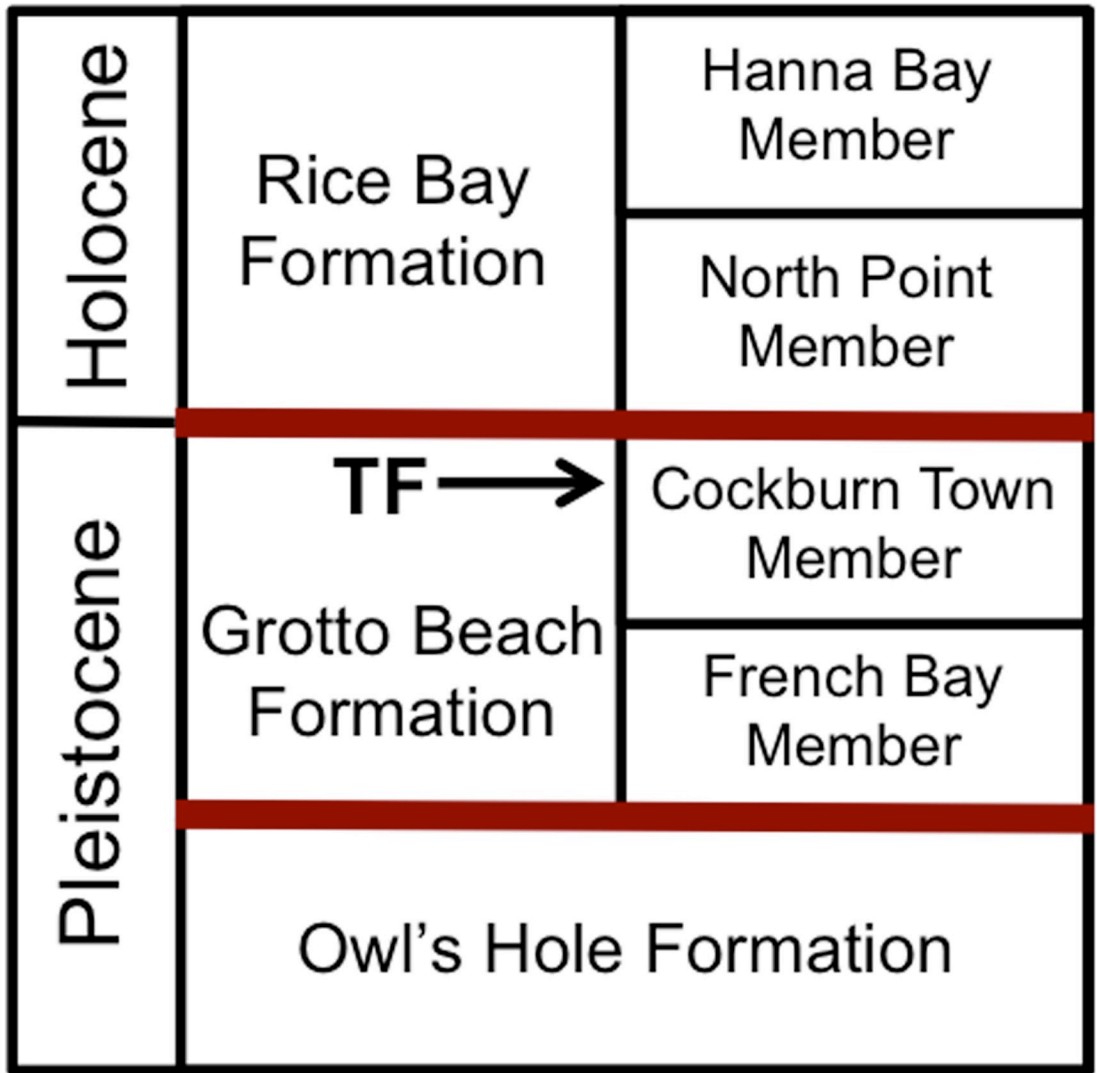

**Fig 2. Pleistocene-Holocene stratigraphy of The Bahamas.** Stratigraphic position of trace fossil (TF) in this study indicated (arrow) toward top of Cockburn Town Member of Grotto Beach Formation, estimated age of 115 *kya* for the trace fossil. Stratigraphic terminology after [31,32], with Grotto Beach Formation bounded by paleosols, as indicated by red lines. Formation and member thicknesses not to scale.

planes, but these are far less common than root trace fossils (Fig 4C). During our 2018 visit, one of us (MMH) also discovered a larger vertical burrow in the outcrop about 5 m west of the main trace fossil of interest in this study (Fig 4D). This burrow is expressed in full relief and filled with sediment identical to that in the surrounding lithology. Its top portion is located about 90 cm below the micritic layer. The burrow is tubular, 52 cm long, and 4–5 cm wide, but with two thicker sections that spiral around a central axis near its base. We found no fossil burrows in the outcrop attributable to ocypodid crabs (e.g., *Psilonichnus*), which are typical of coastal eolianites on San Salvador and elsewhere in The Bahamas [40, 43–45].

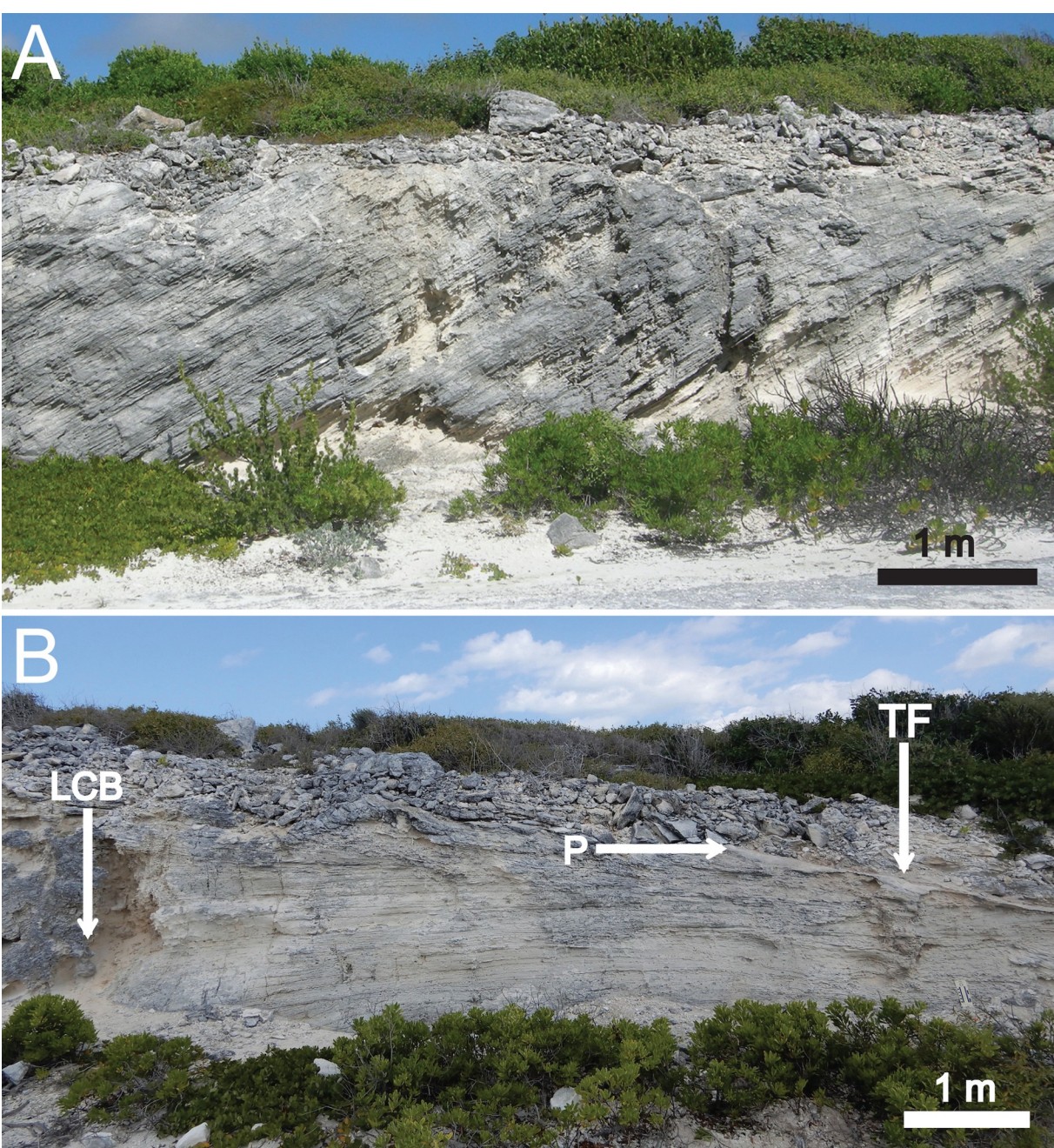

**Fig 3. Outcrop of Cockburn Town Member (Grotto Beach Formation) eolianite at study site, San Salvador Island, The Bahamas.** (A) Middle portion of outcrop, with high-angle eolian cross-bedding partially disrupted by vertical and oblique root trace fossils and weathered rubble on top. Scale bar = 1 m. (B) Eastern portion of outcrop with stratigraphic position of protosol (P), interpreted fossil land-crab burrow (LCB) and trace fossil that is the main focus of this study (TF). Scale bar = 1 m.

### Trace fossil of interest

The top of the structure is positioned 50–60 cm below the uppermost surface of the outcrop, with the outcrop 1.6 m thick there (Fig 5). The structure stands out in bas-relief with an overall conical outline opening upward. This relief, however, is uneven and protrudes only 1–10 cm from the main face of the outcrop. Nonetheless, the structure is also discernable as a significant

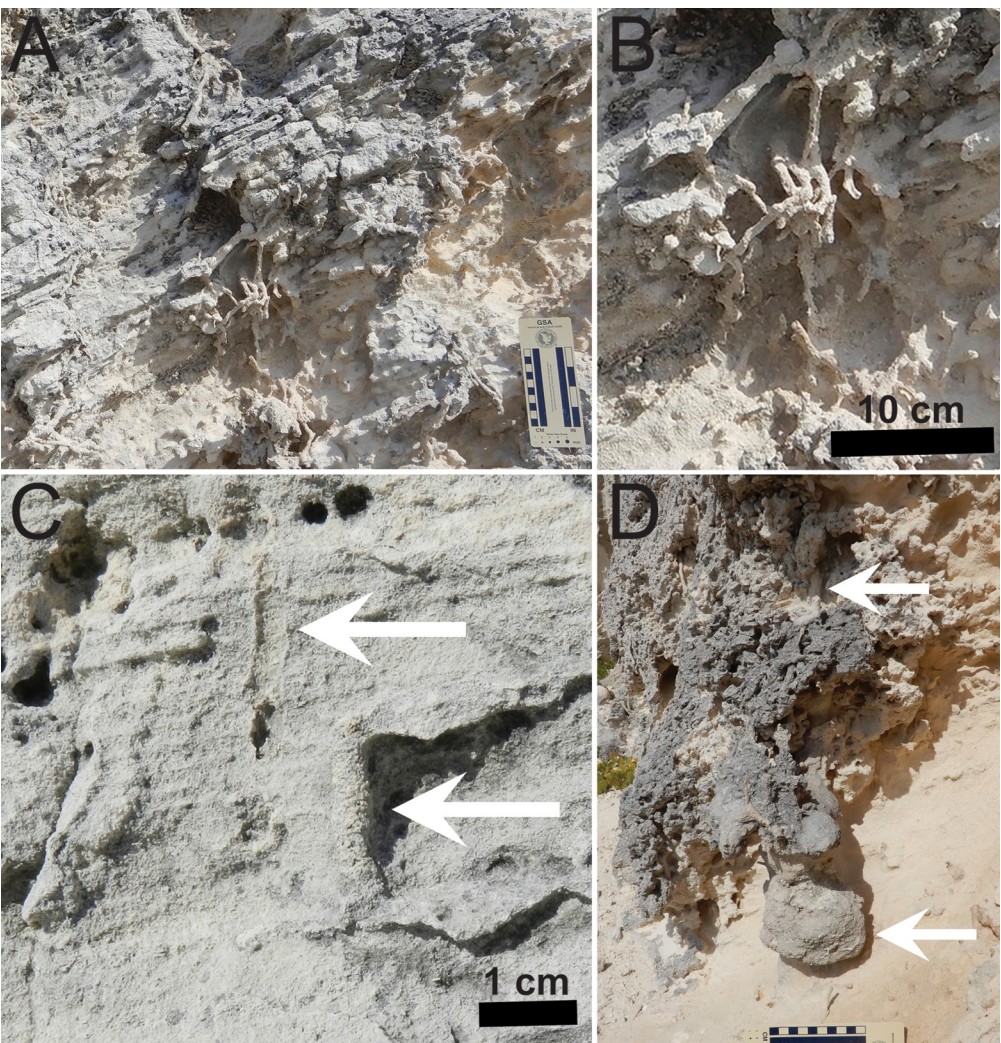

**Fig 4. Plant and invertebrate trace fossils in Cockburn Town Member eolianite at study site, San Salvador Island, The Bahamas.** (A) Branching root trace fossils in high-angle cross-bedding. Photo scale in centimeters. (B) Close-up of root trace fossils. (C) Small vertical burrows (*Skolithos*) indicated by arrows, left expressed as external mold and right as full-relief internal mold. (D) Large vertical burrow (upper arrow) associated with root trace fossils, with thicker, spiraled portion in lowermost portion (lower arrow). Photo scale in centimeters.

vertical disruption of surrounding cross-bedding, with soft-sediment deformation apparent in a 5–10 cm thick zone around it. The structure is capped by the previously described thin micritic layer, which in turn is overlain by cross-bedded oolitic grainstone. This micritic layer dips downward a few centimeters into the structure and defines its uppermost surface.

The straight-line vertical extent of the structure is 69 cm, defined by the distance from its topmost surface–where it meets the micritic layer–to its bottommost end. However, when viewed from above, the structure turns clockwise (dextrally), lending a subtle "J" shape that is 82 cm long when measured along this spiraling length. The structure has a maximum width of 40.5 cm at its top, defined by a gentle sloping surface on its left and an abrupt vertical surface on its right, with both cutting across bedding. Widths in its middle vary from 19 to 26 cm. The higher-relief part of the structure narrows downward to 12 cm wide near its base, and is about 5 cm wide at its sub-rounded end. However, an eroded portion of the structure extends about

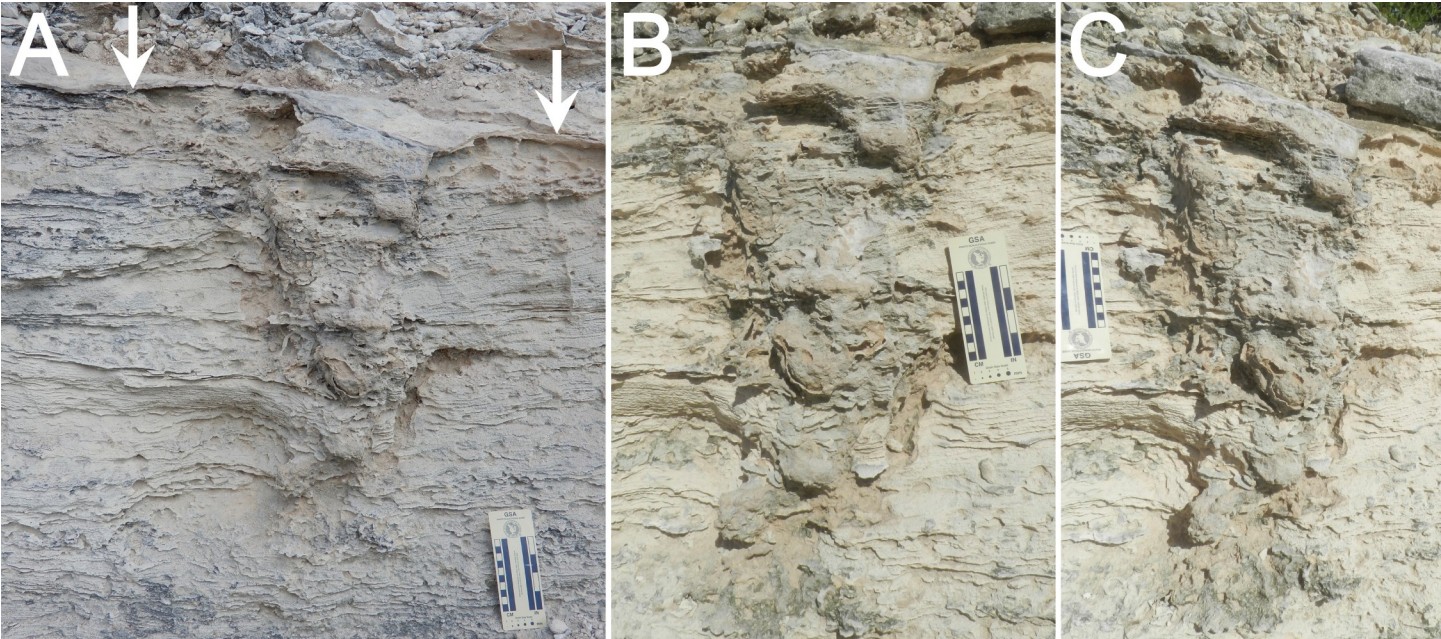

**Fig 5. Fossil iguana nesting burrow, San Salvador Island, The Bahamas.** (A) Overall longitudinal view of structure, with protosol indicated (arrows). (B) Right-side oblique view. (C) Left-side oblique view. Scale in all photos in centimeters.

10 cm below and 5 cm to the right third of the main structure; this part is more discernable as disruption of surrounding bedding rather than relief.

The top surface above the main vertical trend of the structure is slightly concave and approximates an ellipse in outline, with widths varying from 28–33 cm. Below this top surface, the structure has internal layering punctuated by at least six curved (concave upward), thin (2–12 mm) compacted zones bounded by irregular lower surfaces (Fig 6). The uppermost five zones are evident as lobate bulges with greater relief than surrounding sediment and are vertically separated at semi-regularly spaced intervals of 9–12 cm below the top of the structure. The narrow, basal high-relief part is similarly compacted, whereas the lower-relief part of the structure below is not compacted and vaguely defined. This basal part is flat-bottomed and about 20 cm wide.

We saw no evidence of dissolution and/or precipitation of calcitic layers or other minerals along boundaries of the structure. Moreover, concentrations of hematitic minerals typically associated with epikarst in The Bahamas [31, 39, 46] are limited, with infrequent or subtle orange colorations within and just outside of the structure. The lithology outside of the structure is composed of very fine to medium oolitic sand, differing slightly from inside, which consists of fine-medium sand with a concentration of coarse sand and granules in its top 10 cm. Lateral boundaries of the structure and other parts of it are cross-cut by root trace fossils that continue partially into the interior. Root trace fossils are also apparent 10–20 cm outside of the main structure in the outcrop (Fig 7). In eroded parts of the structure, bedding is discernable behind it in the outcrop, but is discontinuous within.

## Discussion

We interpret the main structure of interest as a biogenic sedimentary structure, and specifically as a vertebrate burrow formed originally on the leeward flank of an eolian dune. More explicitly, this trace fossil is consistent with the size, depth, and internal structure of a nesting

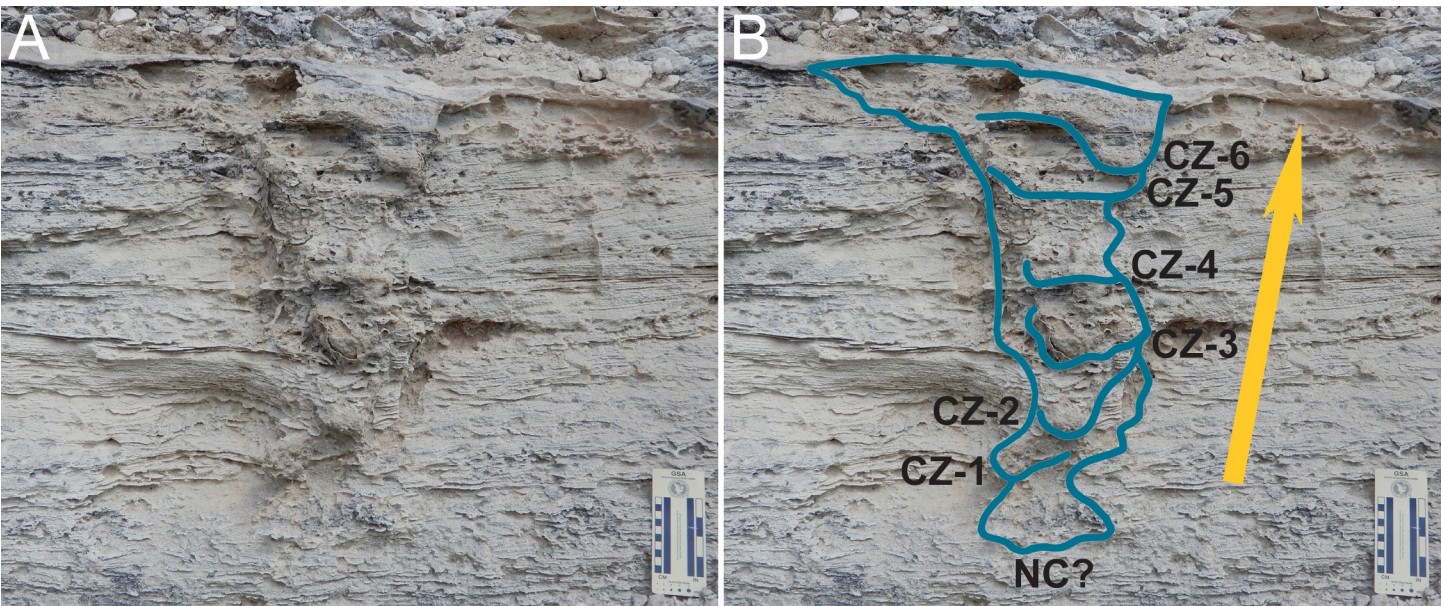

**Fig 6. Outline of fossil iguana nesting burrow with compacted zones.** (A) Full view of nesting burrow, with bas-relief expression on left and center, and low-relief expression in its bottom-right, cutting across and in front of surrounding cross-bedding. (B) Labeled outline of burrow with possible nest chamber (NC?) at base and compacted zones (CZ) indicated, indicated in order of probable formation by the tracemaker as it moved from bottom to top and back-filled burrow, CZ-1, CZ-2, and so on; arrow indicates direction of movement by tracemaker as it backfilled the burrow. Scale in centimeters.

burrow made by an iguana, such as those of various species of modern *Cyclura* native to The Bahamas and the Caribbean [21–24, 47]. Its facies setting also reflects a proper nesting environment for iguanas, which build nests in terrestrial settings, including near-coastal dunes [18, 19, 21–24, 47, 48]. We considered alternative explanations for this structure and falsified each based on criteria described herein.

## Paleoenvironmental setting

The overall geometry of the outcrop and its cross-road counterpart, as well as the lithology (oolitic grainstone), high-angle cross-bedding, root trace fossils, and lack of marginal-marine trace fossils, e.g., ghost-crab burrows [40, 41, 43–45], summarily indicate that the original sedimentary environment was a near-coastal dune: not a primary coastal dune directly adjacent to a beach, but more inland (Fig 8A). Our diagnosis of the sedimentary environment affirms previous interpretations of this and nearby outcrops as regressive eolianite facies of the Cockburn Town Member within the Grotto Beach Formation [30, 31, 35, 36]. Facies at this outcrop were more specifically identified as part of a lobate dune ridge formed and preserved through rapid accumulation and cementation, respectively [35]. Eolianite facies here were likely formed as regressive dunes during a Pleistocene sea-level drop following the Stage 5e sea-level high [30, 31, 36, 37]. Assuming this deposit and the corresponding outcrop south of it composed the same former dune, the trace fossil is located on its northeastern flank.

The thin micritic layer overlying the structure is likely a poorly developed (immature) paleosol, or protosol; protosols are common in eolianite facies of San Salvador and other islands of The Bahamas [31, 35, 37, 39, 49]. Interpretation of this micritic layer as a protosol is supported by its near-continuous lateral extent throughout the outcrop, signifying a surface that was briefly exposed to subaerial soil-forming processes before burial by overlying dune

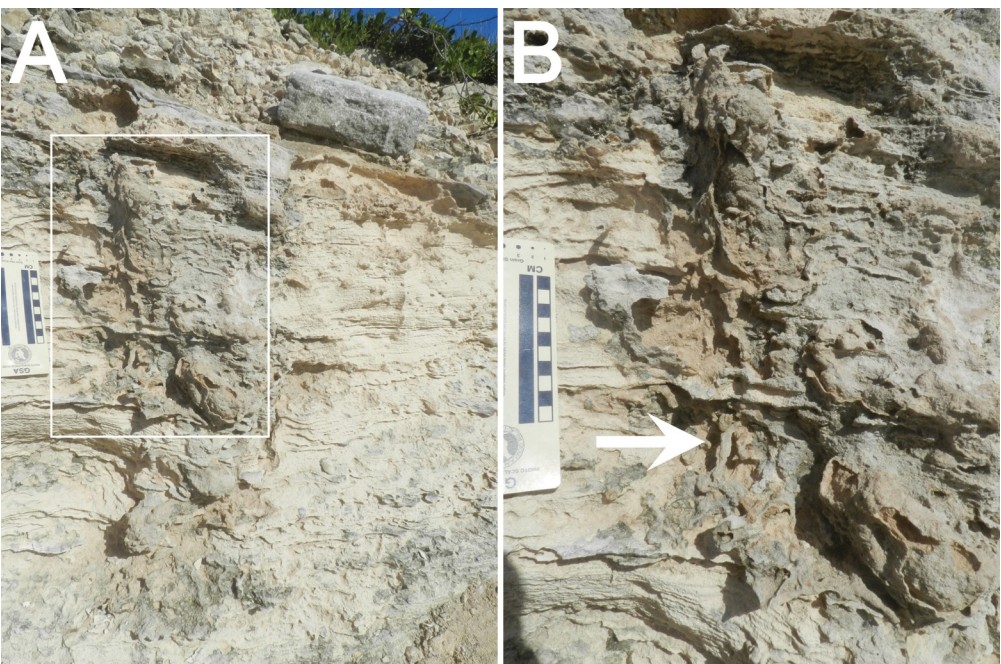

**Fig 7. Boundary of fossil iguana burrow and root trace fossils.** (A) Left side burrow boundary showing diffuse, soft-sediment deformation of surrounding bedding and root trace fossils (inset). (B) Close-up of boundary with root trace fossils cutting across boundary (arrow). Scale in centimeters.

sediments [35]. Root trace fossils in the outcrop (Fig 4A and 4B), including some directly associated with the structure boundary (Fig 7), likewise point toward sufficient time for terrestrial plants to have colonized dune surfaces, and more so toward the dune crest [35, 42, 43]. Less common are thin vertical structures (Fig 4C), which we interpret as insect or arachnid burrows; such trace fossils also have been reported from eolianites on San Salvador and other islands in The Bahamas [40, 41, 43, 44]. However, during our 2018 visit to the outcrop, we also noted a prominent vertical structure about 5 m west of the interpreted iguana burrow (Fig 4D). Because of its relatively wide diameter (4–5 cm), depth, and setting in an inland eolian dune, we interpret this vertical structure as a land-crab burrow, similar to those made by an adult black-backed crab (*Gecarcinus lateralis*) or a juvenile blue land crab (*Cardisoma guanhumi*) [50–53].

The lithology, physical sedimentary structures, and associated trace fossils therefore indicate fully terrestrial conditions for this dune deposit, although most of its sediments were likely derived from nearby coastal environments. Oolites in particular would have been supplied by a formerly nearshore carbonate platform south of San Salvador exposed during the Stage 5e regression [30, 31, 35, 39]. Such a paleoenvironmental setting would have provided suitable habitats for burrowing iguanas, whether for dwelling or nesting. For example, modern species of *Cyclura*, such as *C. nibula* of Cuba, often burrow in vegetated dunes near beaches, as opposed to other habitats, such as xerophytic coastal scrub [47]. Similarly, *C. rileyi rileyi* digs nests in sandy substrates next to beaches, although this species also nests in sandy or rocky island interiors [22, 24]. The disruption of primary cross-bedding by the trace fossil, inclusion of coarser-grained clasts in the sediment fill near its top, and its position directly below an interpreted protosol implies the burrow was excavated at an immature soil surface. This is again consistent with its origin as a burrow made in terrestrial conditions.

## Tracemaker identity and behavior

The width of the topmost surface (~40 cm) of the trace fossil and throughout most of its length (19–26 cm) overlaps with burrows widths of *Cyclura* spp. in The Bahamas and Caribbean [17, 21, 22, 26, 47]. The vertical depth (69 cm) and overall length (82 cm) of the burrow further coincides with reported depths and lengths of burrows made by species of *Cyclura* and related iguanas; its J-shape also matches that of burrows made by iguanas that turn either right or left with descent [17, 19, 21, 22, 26, 47]. Although the top surface does not offer a clear orientation of the burrow, the cross-sectional view of the structure on an east-west trending outcrop implies that it would have been oriented north-south, turning west with its deepest portion. Moreover, the gentle sloping surface on its left may represent part of the original ramp leading down into the main burrow, which would imply its entrance was from the northwest. However, the making of the original roadcut also likely destroyed much of the original structure, rendering its full geometry and orientation uncertain.

The internal structure of the burrow, with semi-regularly spaced compacted zones, indicates that its tracemaker actively backfilled it. These compacted zones identify the structure as an iguana nesting burrow, rather than a dwelling burrow, which would have remained open unless collapsed or otherwise filled passively (Fig 8B). Female iguanas of every modern species are responsible for making nesting burrows, which they construct by digging with all four limbs [17–19, 21, 22]. Once a female iguana selects a nesting site, she displaces sediment out of the initial burrow to form an apron, which becomes more voluminous with progressive movement down and into the host substrate [17–19, 21, 22]. However, initial digging does not always end with a nesting burrow. Female iguanas may first make exploratory burrows, but if they regard a site as unsuitable, they abandon a burrow after 30–60 cm of digging [21, 22]. Once a burrowing iguana reaches a depth deemed adequate for egg incubation, she turns right or left before making the egg chamber, forming an inclined J-shaped burrow [18, 19, 21]. Tunnels in nesting burrows of some iguana species may also have short branches, or turn enough at their ends to define an inclined U-shape [21]. Closely spaced and cross-cutting burrows can even form burrow complexes, in some instances dug by two species nesting in the same area [18, 20].

If an iguana does not abandon a nesting burrow before completing it, then its bottommost portion corresponds with an egg (nesting) chamber. In contrast to sea-turtle nests or egg chambers made by varanids, an iguana nesting chamber may be just slightly wider than the main tunnel, only allowing enough room for the mother iguana to turn around [17, 21, 22]. Egg clutches for species of *Cyclura* in The Bahamas and parts of the Caribbean range from 1–24, but average about ten [3, 21, 22, 54], whereas clutches of other species of iguanas are greater. For instance, green iguanas (*Iguana iguana*) can produce clutches of more than forty eggs [17, 19]. Loose sediment normally collapses behind a burrowing iguana as she moves deeper toward the egg chamber, meaning she is buried alive while laying eggs; afterwards, she may stay in the nesting burrow for more than a day to protect the clutch [19, 22]. Iguanas then exit headfirst, pushing sediment behind them and otherwise packing the burrow with head and limbs [17, 19, 21, 22]. Such behaviors should produce thinner, compacted layers separating looser sediments. Once at the surface, nesting females kick much of the sediment apron into and over the top of the burrow to cover it, forming a low-profile mound [17, 21, 22]. Expectant mother iguanas then normally stay near their concealed nesting burrows and guard them until the eggs hatch, which for different species of *Cyclura* can range from 63 to 119 days [21 and references therein; 54]. For *C. rileyi riyeli*, incubation is about 90 days [22]. Hatchlings then must dig out of the nesting burrow to reach the surface [19, 21, 22].

Although the trace fossil diagnosed here is an incomplete expression of the original full-sized burrow, its width, depth, overall form, and internal structures are nonetheless consistent

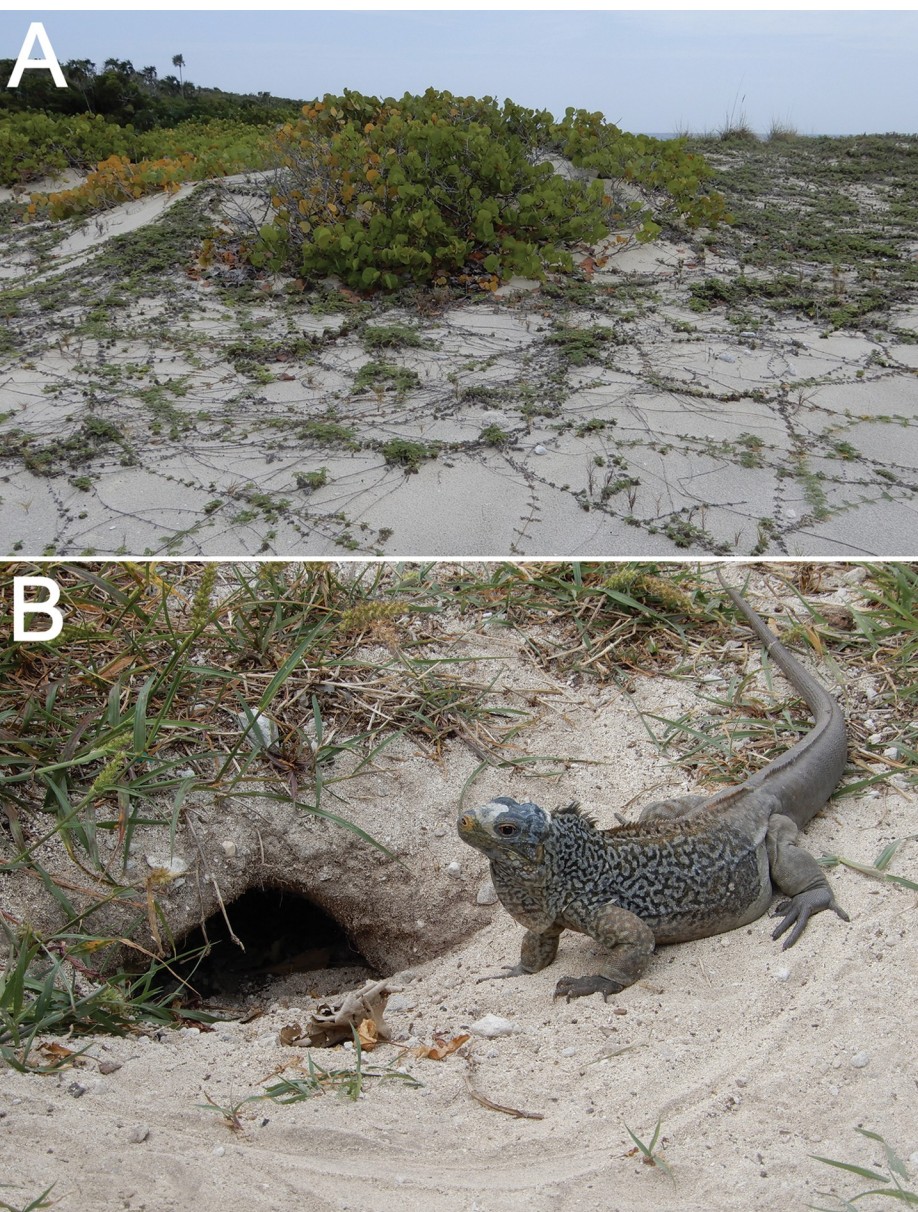

**Fig 8. Modern analogs of interpreted nesting environment and tracemaker for facies and trace fossil, respectively.**
(A) Inland vegetated dune, mostly colonized by *Coccoloba uvifera* (sea grape) and composed of oolitic-skeletal sand, located about 120 m from intertidal zone at Sandy Point, San Salvador Island, The Bahamas. (B) Adult San Salvador rock iguana (*Cylcura riyeli riyeli*) in front of open dwelling burrow at Gerace Research Centre, San Salvador Island (The Bahamas).

with those of modern iguana nesting burrows. For example, in one study of nesting burrows made by the Allens Cay iguana (*Cyclura cychlura inornata*), burrows were 149 +/- 55 cm long (n = 74), with a range of 54 to 330 cm; vertical depths to bottoms of egg chambers were 27.7 +/- 8.2 cm (n = 75), but with a range of 11 to 59 cm [21]. Given a vertical depth of 69 cm for the burrow (as measured from the protosol horizon), the trace fossil is slightly deeper than the maximum depth recorded for nesting burrows by *Cyclura cychlura inornata*, but only by 10 cm. Nevertheless, the San Salvador trace fossil does not exceed maximum vertical depths

recorded for nests of other species of *Cyclura* [21], with 76 cm reported for *C. cornuta stejnegeri* [55], 90 cm for *C. pinguis* [56] and 70 cm for *C. nubila nubila* [57]. Granted, the complete original length of the fossil burrow is indeterminable, as it may have started at a higher surface than the protosol.

The minimal 82 cm length throughout the descent of the burrow is likewise within the range of burrow lengths reported for nearly all species of *Cyclura* and similar to those of burrows made by modern San Salvador rock iguanas (*C. rileyi rileyi*) on San Salvador and nearby cays [21, 22]. Moreover, this added length from the J-shaped form of the burrow accords with those of iguana nesting burrows [18, 19, 21]. Although the fossil burrow may have an approximate orientation dipping to the south, and possibly with a northwest-facing entrance, *Cyclura* nesting burrows are not known to have any preferred orientation [21, 22, 26]. Hence the burrow trend in this specimen may not reflect any knowable behavioral significance.

The coarser sediment fill inside the fossil burrow with respect to the surrounding eolianite lithology further suggests its maker actively filled it. This fill would have resulted from the tracemaker breaking up the adjacent partially cemented surface soil. This action would have formed lithoclasts, which were then included in the sediment fill with further digging. Additional features supporting an actively back-filled burrow are the six regularly spaced, compacted zones within the burrow fill (Fig 6), which we interpret as an internal structure made as its maker exited the burrow. This backfilling episodically compacted sediment within the burrow, forming large-scale meniscae that also were more resistant to weathering in the outcrop, thus imparting higher relief. Such periodic compaction within a burrow and its resultant patterns are analogous to actively backfilled invertebrate burrows, exemplified by trace fossils such as *Taenidium* and *Ancornichnus* [58, 59]. However, a few modern vertebrates, such as marsupial moles, also actively back-fill their burrows [60]. Compaction was likely accomplished by a combination of the iguana using her forelimbs and head, as observed in modern iguanas [17, 21, 22]. As a result, the compacted layers within the sediment fill of the fossil burrow are consistent with this behavior. In contrast, passively filled burrows–whether done by water- or wind-borne sediments, collapse of the burrow, or a combination of these–should result in more chaotic arrangements of sediments and lack regularly spaced compacted layers. Furthermore, if the burrow were dug as an incomplete exploratory "nest" lacking an egg chamber, it would not have been actively backfilled, either [17, 21, 22]. In summary, because of this evidence for an active fill and sediment packing in the burrow, we correspondingly conclude the San Salvador trace fossil is not only an iguana burrow, but also one made as an active nest by a nesting female. Also, because only female iguanas make nesting burrows, this trace fossil is a rare example of how the tracemaker's sex is interpretable.

Still, if this is a nesting burrow, it is probably that of a failed nest. For instance, we did not see secondary disruptions of compacted layers in the exposed surface of the burrow. Such disruptions are an expected result of hatchlings digging up through sediments from the nesting chamber to the surface [19, 21, 22]. More evidence suggesting a nest is in the lowermost and lower-relief portion of the burrow, which widens to about 20 cm. This part may represent the terminal egg chamber, or at least marks where the mother iguana turned upward to exit. However, any embryonic iguana remains or eggs would have disintegrated since their emplacement, and this end does not bear any clear evidence of external or internal molds of eggs. Yet if this bottom portion does represent an egg chamber, its relatively small size also implies a small clutch size, which is more akin to modern *Cyclura* species (less than 10 eggs/clutch) versus *Iguana* species (more than 40 eggs/clutch) [19]. Our interpretation is admittedly speculative, though, and would require testing via comparison of the San Salvador structure to similar trace fossils, or casts made from failed modern iguana nesting burrows.

The burrow cut through well-defined cross-bedding in the dune, and hence made a disturbed zone in sediments conducive to plant colonization. This supposition is supported by root trace fossils concentrated along the edge of the burrow (Fig 7), perhaps contributed by plants near the burrow entrance that colonized the incipient soil after the burrow was made. Modern iguanas commonly burrow in bare, sandy areas, but also use vegetated areas as cover for both dwelling and nesting burrows [17, 19, 21, 22, 47, 54].

## Alternative hypotheses for structure

Alternative hypotheses for the structure are that it is: (1) a soft-sediment deformation structure (collapse) formed by physical or biogenic processes; (2) an erosional structure produced by post-depositional solution of the limestone (e.g., epikarst); or (3) a burrow made by an animal other than an iguana. In the following analysis, we tested and falsified each of these alternatives, leaving an iguana nesting burrow as the remaining explanation.

Eolianites from other locations and geologic ages contain soft-sediment deformation structures formed by non-biogenic processes, such as liquefaction and fluidization, with some caused by seismic activity [61, 62]. However, such features have not yet been reported from Pleistocene or Holocene eolianites of San Salvador [35]. The Bahamas Platform also has been tectonically stable during much of its geologic history [31, 63], although southern islands more proximal to the Caribbean plate margin are likely uplifted [64]. Accordingly, seismites would be unexpected on San Salvador, and have never been reported there. Relatively large animals walking on dune slopes can also cause collapse structures in dunes by starting down-slope grainflows. Such structures, though, are much wider than deep, and do not extend into dunes as conical features [65]. We also are not aware of any animals living on San Salvador during the Late Pleistocene that were large enough to cause dune collapses of this magnitude with their footsteps.

We likewise reject epikarst as an explanation for the structure as it lacks features typically associated with dissolution, secondary precipitation, and hematite concentrations associated with paleosols (*terra rosa* horizons) in The Bahamas [31, 37, 39]. Boundaries of the structure are relatively diffuse, rather than sharply defined, with the latter more typical of dissolution of lithified sediment and re-precipitation. Although the structure contains some orange hues attributable to hematite–indicating oxidation of iron-bearing minerals within the structure–these are not nearly as concentrated as in epikarst and most paleosols of The Bahamas [31, 37, 39]. Hematite in Bahamian paleosols is connected to iron-rich clays and other minerals in wind-blown dust derived from the Sahara Desert, which is particularly concentrated during sea-level lows [31, 32].

We also considered that the trace fossil might have a plant origin, whether as a trace fossil representing the lower part of a root system or the body fossil of a relatively small tree trunk. Similar large-diameter structures attributed to trees are preserved in Pleistocene eolianites on other Bahamian islands and Bermuda [66, 67]. However, the internal structure of the trace fossil, consisting of multiple compacted layers (Fig 6), argues against passive filling of either a decayed or otherwise vacated root structure. Similarly, the structure shows no evidence of external molds from a trunk or other plant parts [66]. Although root trace fossils are along the edges and outside of the structure, these are apparently secondary, as they cut across the deformational boundary of the structure (Fig 7). These trace fossils may represent plant colonization coinciding with the formation of the protosol capping the structure. However, root trace fossils well outside of the structure and likewise cutting across primary bedding suggest several generations of plant colonization concomitant with soil formation and stabilization of the dune [35].

Assuming the structure is a burrow, then an animal other than an iguana may have made it. Of these, only a few probable Pleistocene burrowing invertebrates or vertebrates in The

Bahamas can be considered as potential makers. Size, however, is a limiting factor for invertebrate tracemakers. Of invertebrates native to The Bahamas and likely living on the Bahamas Platform during the Pleistocene, the blue land crab (*Cardisoma guanhumi*) is the only one capable of making burrows approaching the width and depth of this structure. This decapod's open burrows can be as wide as 18 cm, as long as 3–4 m, and reach maximum depths of 1.5 m; depth is influenced by local water tables, which crabs attempt to reach with their burrows [50]. These open burrows have one or two entrances with a simple, unbranched helical tunnel that descends from nearly horizontal (10˚) to nearly vertical (75˚) [50, 51]. Individual tunnels may have short branches or intersect underground, but all have rounded chambers at tunnel ends, and are not actively back-filled [50, 53]. In contrast, the structure described in this study is wider than 20 cm throughout most of its length, has one only apparent entrance, is not helical, does not terminate with a rounded chamber, and was likely backfilled. The fortuitous discovery of a probable land-crab burrow near the structure further aids in our contrasting its characteristics with the interpreted iguana nesting burrow (Fig 4D). Also lacking in the burrow are the shafts, tunnels, chambers, and other traits of a social insect nest, such as those made by ants, termites, wasps, or bees [68]. Accordingly, we reject an invertebrate origin for this trace fossil.

Late Pleistocene vertebrates in The Bahamas other than iguanas capable of making a burrow of this size and form are also limited. Nonetheless, some to consider include: sea turtles, such as loggerheads (*Caretta caretta*), hawksbills (*Eretmochelys imbricata*), green turtles (*Chelonia mydas*), and others; land tortoises (*Chelonoidis alburyorum*); the Cuban crocodile (*Crocodylus rombifer*); and ground-nesting birds, such as species similar to the modern Audubon shearwater (*Puffinus lherminieri*).

Female sea turtles dig out nests along sandy shorelines of The Bahamas today, and likely made nest structures in coastal dunes during the Pleistocene. Sea-turtle nests are excavated on or just behind primary dunes of shorelines, often overlapping with the ecological range of ghost-crab burrows [69]. Such nests are complex traces, consisting minimally of: crawlways produced by the mother turtle; a broad covering pit; a turtle-sized body pit; and a vertically oriented subspherical egg chamber [69, 70]. Turtles use rear flippers for digging out a nest and burying the egg clutch, which is covered with loose sand [69–72]. If a clutch successfully hatches, hatchlings also disturb overlying sediments of the egg chamber and the body pit. Hatchlings that breach the surface generate dozens of horizontal crawlways leading from the nest. Nest depths and widths vary with species, but are well documented for loggerheads, with body-pit depths of about 20 cm and egg-chamber depths of 35 cm [70]. None of the preceding qualitative traits of sea-turtle nests are present in the San Salvador structure, and its depth exceeds that of any known modern sea-turtle nest. Additionally, its facies setting differs from typical sea-turtle nests, which occur in near-coastal dunes, forebeaches, and washover-fan deposits [70, 73]. Sea-turtle nests are accordingly co-associated with ghost-crab burrows in near-coastal dunes [69], whereas the outcrop hosting the San Salvador structure lacks trace fossils attributed to ghost crabs, such as *Psilonichnus upsilon* [40, 41, 43–45]. Again, a lack of ghost-crab burrows implies this dune was more inland, and not coastal.

Land tortoises were other possible makers of large burrows in Pleistocene sediments of The Bahamas, although these were not likely to have produced trace fossils resembling the one described in our study. Fossils and subfossils of a giant land tortoise, *Chelonoidis alburyorum*, have been reported from the northern part of The Bahamas [74, 75], but are unknown on San Salvador [30]. Although some tortoise species burrow, such as gopher tortoises (*Gopherus polyphemus*) of the southeastern U.S., such excavations are open dwelling burrows [69, 76, and references therein]. Gopher tortoises and other tortoise species dig shallow, horizontally oriented nests in sand aprons outside of their dwelling burrows [69, 76], which also does not match the structure described here.

The Cuban crocodile (*Crocodylus rhombifer*) is another large vertebrate that may have made nests in near-coastal dunes. Based on fossil and subfossil remains, as well as post-Colombian historical records, *C. rhombifer* was widespread in The Bahamas during the Pleistocene and Holocene, including San Salvador [77–80]. Female Cuban crocodiles bury their eggs, but unlike some other crocodilian species, they do not make hole nests. Instead, they dig either trenches or construct mound nests [81]. Furthermore, nests are normally made near freshwater marshes and covered with plant matter, rather dug into coastal dunes or other sandy substrates [81]. Again, the form, substrate, and probable habitat of the San Salvador structure are inconsistent with those that would be expected for a fossil crocodilian nest.

Of burrowing birds native to The Bahamas, the Audubon shearwater (*Puffinus lherminieri*) or similar species also could have constructed burrows with widths comparable to that of the San Salvador structure. Nevertheless, their burrows differ in key traits. Shearwaters use burrows as nests and for protecting hatchlings until they fledge [76, 82–84]. Shearwater burrows are, however, kept open throughout their use, and consist of simple tunnels measuring as long as 3 m [76, 83, 85]. Shearwaters and other burrow-nesting birds also do not backfill their burrows, but keep these open [76, 83]. Thus these burrows are only filled through collapse or otherwise passive means, which again rules out the San Salvador structure as a bird burrow.

In summary, after comparing the San Salvador structure to non-biogenic structures and biogenic structures of other possible tracemakers, and eliminating these as possible explanations, we conclude it is the trace fossil of a nesting iguana burrow. We acknowledge that the trace fossil may represent some previously unknown natural phenomenon, but its close resemblance to modern iguana burrows, its same paleoenvironmental setting, geographic co-occurrence with a species of burrow-nesting iguana species, and relatively young geologic age all support our interpretation. Considering that only one species of *Cyclura* does not make nest burrows [3, 23, 48, 86], and that San Salvador hosts a modern nest-burrowing species (*C. rileyi rileyi*), we suggest that future researchers compare nest burrows of this and related species in The Bahamas to the San Salvador trace fossil to further test our interpretation. We also hope that it will serve as a model for identifying similar structures that may be preserved in the widely exposed Pleistocene-Holocene eolianites of The Bahamas or in the Caribbean.

## Conclusions

The trace fossil described here from the Pleistocene Grotto Beach Formation (~115 *kya*) of San Salvador Island is the first known fossil iguana nesting burrow interpreted from the geologic record. Its size, form, paleoenvironmental setting, and location on an island with a native species of iguana (*Cyclura riyeli riyeli*) further support its identity as a fossil iguana nesting burrow; body fossils of *Cyclura* are also in Pleistocene deposits of The Bahamas. Because only female iguanas make nesting burrows, this trace fossil serves as an uncommon instance of when the sex of a tracemaker is interpretable. Although we cannot state definitely whether the burrow was made by a species of *Cyclura*, its dimensions and form are consistent with nesting burrows made by modern species of *Cyclura* currently on San Salvador and nearby cays [21, 22, 24, 28, 29]. This fossil iguana burrow also establishes a minimum time for iguanas on San Salvador during the Middle-Late Pleistocene, which can be applied to better understanding the biogeographic dispersal of *Cyclura* throughout The Bahamas during the Pleistocene and Holocene [87–89]. Our hypothesis can be tested by either the absence or presence of more such trace fossils in eolianites of San Salvador and elsewhere in The Bahamas, perhaps associated directly with body fossils of their tracemakers [*sensu* 90, 91].

Of more local significance, this trace fossil is only the second attributed to vertebrates on San Salvador Island, with the first having been two shorebird tracks [92]. Moreover, it is the

only fossil vertebrate burrow interpreted from The Bahamas. Nevertheless, vertebrates other than iguanas that are native to The Bahamas living on or otherwise intersecting with terrestrial environments–such as sea turtles, tortoises, and shearwaters–should have also left trace fossils of their presence in the Pleistocene-Holocene eolianites there. It may be simply a matter of looking for such trace fossils to increase our sample size.

## Supporting information

**S1 Fig.**
(TIF)

**S1 File.**
(DOCX)

## Acknowledgments

We thank *PLOS One* associate editor, Dr. David Lovelace, for his editorial steering during and after the review of this manuscript, and for helpful feedback provided by two anonymous reviewers, which collectively contributed to our presenting a more organized report. The senior author (AJM) appreciates the long-time and reliable logistical support of the Gerace Research Centre (GRC) on San Salvador Island. In particular, we want to thank Tom Rothus, who was director of the GRC when two of us (AJM and MW) first noticed and described the trace fossil in 2014, and the current director, Troy Dexter. The Emory University Office of International and Summer Programs organized travel arrangements for four of us (AJM, MW, DS, and MMH) in our teaching field courses on San Salvador. We thank Emory University undergraduate students in the 2013–14, 2015–16, and 2018 field courses who witnessed our discovery and some of the subsequent scientific investigations of this trace fossil, an example of where teaching contributes to research and vice versa. Chase Lovellette (Emory Center for Digital Scholarship) helped set up links to supplemental files. We are always grateful to the people of San Salvador Island specifically and the people of The Bahamas in general for their hospitality to those of us who are mere visitors to their beautiful places with their unique natural histories. Lastly, we wish to acknowledge that this research was conducted on an island originally called Guanahani, which was part of the ancestral range of the Taino people and is now home to descendants of formerly enslaved Africans. Surely among their ancestors were those who observed nesting behaviors of modern rock iguanas well before our study of this trace fossil. We are just now catching up.

## Author Contributions

**Conceptualization:** Anthony J. Martin, Dorothy Stearns.

**Investigation:** Anthony J. Martin, Dorothy Stearns, Meredith J. Whitten, Melissa M. Hage.

**Methodology:** Anthony J. Martin.

**Visualization:** Michael Page, Arya Basu.

**Writing – original draft:** Anthony J. Martin, Dorothy Stearns.

**Writing – review & editing:** Anthony J. Martin, Dorothy Stearns, Meredith J. Whitten, Melissa M. Hage, Michael Page, Arya Basu.

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
