## [Decision Letter · Decision Letter 0]

10 Sep 2020

PONE-D-20-22231

First Known Trace Fossil of a Nesting Iguana (Pleistocene), The Bahamas

PLOS ONE

Dear Dr. Martin,

Thank you for submitting your manuscript to PLOS ONE. After careful consideration, we feel that it has merit but does not fully meet PLOS ONE’s publication criteria as it currently stands. Therefore, we invite you to submit a revised version of the manuscript that addresses the points raised during the review process.

Although there is disparity between the 'accept' and 'major revisions' as a review outcome, I believe there is some consistency in the request for organization (addition of sub headings, etc.) to clarify flow and add structure to the 'interpretations' section. I am inclined to agree with reviewer 1 that this is a well done study, and although reviewer 2 suggests that, for the sake of brevity, you might condense the 'play by play' of the study history - I think that this level of detail adds insight to the overall process of the investigation. It supports your methods over a number of years and is very detailed account of your photogrammetric techniques. It would be possible to condense, but I would not say it is required. I will leave that up to you (the authors) to decide if you feel the level of detail is too much, or necessary to the overall description. The fact that there is significant landscape modification with extreme weather events is an interesting component - this locality survived Hurricane Joaquin, but it is certainly susceptible to future storms which could destroy this structure and thus its detailed documentation is warranted and appreciated.

I appreciate the level of detail, and well supported arguments for and against various trace makers. This is a very strong and well written MS, I appreciate your time making it so. Please consider the following and resubmit your revisions. I do not foresee this needed to go back out for review after the points of the reviewers are addressed (none significantly affect the body of research, data, or interpretations).

TLDR: Consider the overall organization.

- At minimum add subheadings as appropriate to the 'interpretations section'

- Reviewer 2 suggests a significant reorganization of the MS; I do not think this level of reorganization is required, but I would appreciate your consideration of their point(s) - I do not wish to dictate how your MS should look, I want the MS to reflect your storytelling, not mine. If you would like it to remain as is, just highlight your reasoning with your rebuttal letter and I will support that.

Minor Points:

You refer to 'root trace fossils' often, but this could be shortened to rhizoliths (sensu Kraus and Hasiotis, 2006: DOI: 10.2110/jsr.2006.052). It is a bit pedandic (no pun intended) but something to consider.

If you have any questions, please feel free to reach out and I will clarify as needed. I look forward to seeing your revisions.

We look forward to receiving your revised manuscript.

Kind regards,

David M. Lovelace, Ph.D.

Academic Editor

PLOS ONE

Journal Requirements:

2.We note that [Figure(s) 1] in your submission contain [map/satellite] images which may be copyrighted. All PLOS content is published under the Creative Commons Attribution License (CC BY 4.0), which means that the manuscript, images, and Supporting Information files will be freely available online, and any third party is permitted to access, download, copy, distribute, and use these materials in any way, even commercially, with proper attribution. For these reasons, we cannot publish previously copyrighted maps or satellite images created using proprietary data, such as Google software (Google Maps, Street View, and Earth). For more information, see our copyright guidelines: http://journals.plos.org/plosone/s/licenses-and-copyright.

1.    You may seek permission from the original copyright holder of Figure(s) [1] to publish the content specifically under the CC BY 4.0 license. 

Reviewers' comments:

Reviewer's Responses to Questions

**Comments to the Author**

1. Is the manuscript technically sound, and do the data support the conclusions?

Reviewer #1: Yes

Reviewer #2: Yes

2. Has the statistical analysis been performed appropriately and rigorously? 

Reviewer #1: N/A

Reviewer #2: N/A

3. Have the authors made all data underlying the findings in their manuscript fully available?

Reviewer #1: Yes

Reviewer #2: Yes

4. Is the manuscript presented in an intelligible fashion and written in standard English?

Reviewer #1: Yes

Reviewer #2: Yes

5. Review Comments to the Author

Reviewer #1: The manuscript "First known trace fossil of a nesting iguana" is well written, well reasoned, well illustrated, and well referenced. Overall the paper is excellent and I find it to be acceptable as is. The data is well presented and does support the interpretations that have been made, that the described sedimentary structure is first a fossil burrow and second was produced by an iguana as a nesting burrow. I appreciate the lengths to which the authors have gone to demonstrate that the structure is not another feature such as soft sediment deformation or a root trace and that alternate tracemakers are not as likely. The evidence and examples presented are convincing. It would be useful to have some additional subheadings to break up the Interpretation section as it contains information on the interpretation of the environment, the nature of the structure itself, the behavior represented, and the tracemaker.

Reviewer #2: The present study is a quite interesting report of an Iguana fossil burrow in The Bahamas. A report of this nature can be of great interest to a wide audience and therefore worth publishing in PLoS ONE. Nevertheless, due to the descriptive nature of the study, that it is mainly based on interpretations of the substrate and burrow descriptions, without being able to reach deeper analyzes to support the proposed hypothesis, I recommend editing the manuscript to be presented in the short communication format. Indeed, most of the information presented along the manuscript can be synthesized, which would lead to a great reduction in its extension. Additionally, more work should be done on the structure of the article which presents some redundant information and sections. For example, two sections are intended to describe the obtained results (“Results” and “Results and Discussions”). I also recommend avoiding the detailed description about what the authors did during the visits to the study site. This information can be presented in the section “Authors contributions”. In the Results section, there is also information that corresponds to the methods, for example, the study site description.

Below, I suggest a potential structure for the article (I hope to help in organizing the information):

1. Introduction

2. Methods

2.1 Study site (explain where the study was made)

2.2 Lithology characterization (explain how it was made)

2.2 Refuge characterization (just explain how it was made)

2.3 Analysis of ecological evidence (evidence of accompanying flora and fauna)

3. Results

3.1 Lithology description (describe the results regarding this topic)

3.2 Refuge characterization (present the qualitative and quantitative description of the burrow)

3.3 Analysis of ecological evidence (results of this analysis)

4. Discussion

4.1 Discuss the evidence supporting that it is an iguana burrow

A table can be used for summarizing and comparing the Iguana burrow versus burrows built by other animals

4.2 Evolutionary and ecological implications

4.3 Implications for future studies (present suggestions for more studies about this topic, considering methodological issues and open questions).

A better structure of the manuscript will allow the results obtained to be highlighted and better evaluated.

6. PLOS authors have the option to publish the peer review history of their article (what does this mean?). If published, this will include your full peer review and any attached files.

Reviewer #1: No

Reviewer #2: No

---

## [Author Response · Author response to Decision Letter 0]

2 Nov 2020

(The following is also in our response letter.)

Reviewer #1 (Reviewer concerns and suggestions in italics, our replies in bold.)

It would be useful to have some additional subheadings to break up the Interpretation section as it contains information on the interpretation of the environment, the nature of the structure itself, the behavior represented, and the tracemaker.

This is an excellent suggestion, so we have added subheadings to the Interpretation section in the following order: Paleoenvironmental Setting, Tracemaker Identity and Behavior, Alternative Hypotheses for Structure.

Reviewer #2 (Reviewer concerns and suggestions in italics, our replies in bold.)

Nevertheless, due to the descriptive nature of the study, that it is mainly based on interpretations of the substrate and burrow descriptions, without being able to reach deeper analyzes to support the proposed hypothesis, I recommend editing the manuscript to be presented in the short communication format.

While confessing that we are unsure of the reviewer’s meaning of “…without being able to reach deeper analyses to support the proposed hypothesis…” we respectfully disagree with the idea of shortening the manuscript into a short communication. Although this study is centered on the description and interpretation of one trace fossil, we felt its scientific importance warranted a detailed and holistic approach, rather than treating it like an isolated object disconnected from its larger context. For example, the identity of this trace fossil as the nest of a female iguana on San Salvador Island (The Bahamas) in the Late Pleistocene minimally touches on the following topics: how to interpret lizard nesting behavior in the fossil record by using trace fossils and sedimentology without accompanying body fossil evidence; the Pleistocene paleobiogeographic history of iguanas in The Bahamas and Caribbean; and how geological responses to sea-level fluctuations during the Pleistocene may have affected that paleobiogeographic history. Our approach also mimics that of vertebrate paleontologists who offer similarly detailed analyses of bones or teeth, but with an added paleoenvironmental framework that we hope serves as a model for interpreting paleontologically significant vertebrate trace fossils.

Additionally, more work should be done on the structure of the article which presents some redundant information and sections. For example, two sections are intended to describe the obtained results (“Results” and “Results and Discussions”).

This is a good point, as we did not intend to use the heading of Results and Discussion after Results. Accordingly, we changed this heading to Discussion, with new subheadings for sections Paleoenvironmental Setting, Tracemaker Identity and Behavior, and Alternative Hypotheses for Structure.

I also recommend avoiding the detailed description about what the authors did during the visits to the study site. This information can be presented in the section “Authors contributions”. In the Results section, there is also information that corresponds to the methods, for example, the study site description.

We argue to keep this detailed description of what most of the authors did during several visits to the study site, which we included to preserve the unique history of the discovery and subsequent documentation of this important trace fossil. For one, our study differs from most paleontological studies that begin with intentional purpose and follow-through. Rather than taking place during a planned expedition, the discovery of this trace fossil in late December 2013 was accidental, occurring when one of us (Anthony J. Martin) was with a group of undergraduate students while teaching a biannual field course. Secondly, the study was done thereafter “on the fly,” with initial measurements and photos taken in early January 2014, then two more short visits to the outcrop that happened once every two years, fit in between teaching duties while on San Salvador Island.

As an example of how undergraduate teaching was an intrinsic part of this study, the study involved two former students from that biannual field course (Meredith Whitten and Dorothy Stearns) who were both present at its December 2013 discovery and then assisted with measurements in January 2014 and 2016, respectively. Then, during our last visit in March 2018, one of the other coauthors, Melissa Hage (who now co-teaches the field course with Martin) found a previously undocumented fossil invertebrate burrow at the outcrop, diagnosed as that of a land crab, the first such trace fossil identified on San Salvador Island. This fortuitous discovery further confirmed our interpretation of the paleoenvironment and showed the value of repeated short visits to the outcrop, albeit done over nearly five years.

Below, I suggest a potential structure for the article (I hope to help in organizing the information):

1. Introduction

2. Methods

2.1 Study site (explain where the study was made)

2.2 Lithology characterization (explain how it was made)

2.2 Refuge characterization (just explain how it was made)

2.3 Analysis of ecological evidence (evidence of accompanying flora and fauna)

3. Results

3.1 Lithology description (describe the results regarding this topic)

3.2 Refuge characterization (present the qualitative and quantitative description of the burrow)

3.3 Analysis of ecological evidence (results of this analysis)

4. Discussion

4.1 Discuss the evidence supporting that it is an iguana burrow

A table can be used for summarizing and comparing the Iguana burrow versus burrows built by other animals

4.2 Evolutionary and ecological implications

4.3 Implications for future studies (present suggestions for more studies about this topic, considering methodological issues and open questions).

Although the reviewer’s suggested reorganization of the manuscript might seem logical from a biologist’s perspective (an inference we make based on the reviewer’s terminology, e.g., “refuge characterization,” “analysis of ecological evidence”), it is not practical for a paleontological study. Such a reorganization would alter the narrative flow of the report, which currently moves from the geological setting (Pleistocene sea-level changes affecting an isolated island, resulting in the paleoenvironmental setting for the nesting) to the behavior of an individual tracemaker (a Pleistocene female rock iguana attempting to nest in that paleoenvironment) to modern analogs (environmental setting and nesting behaviors of extant rock iguanas), and all places between. In short, retrofitting our study to accommodate the reviewer’s non-geological perspective would unnecessarily complicate the manuscript.

Furthermore, our narrative intentionally avoids using tables so readers can have a readable but succinct report of how the reported trace fossil compares to modern iguana nesting burrows and other phenomena. Many of the references cited in the manuscript report nest dimensions and qualitative traits from a number of extant iguana species, and most of those references have tables of those parameters. Hence if readers wish for that additional information, they are welcome to seek it out in those studies. 

Associate Editor (editor concerns and suggestions in italics, our replies in bold)

Although there is disparity between the 'accept' and 'major revisions' as a review outcome, I believe there is some consistency in the request for organization (addition of sub headings, etc.) to clarify flow and add structure to the 'interpretations' section. I am inclined to agree with reviewer 1 that this is a well done study, and although reviewer 2 suggests that, for the sake of brevity, you might condense the 'play by play' of the study history - I think that this level of detail adds insight to the overall process of the investigation. It supports your methods over a number of years and is very detailed account of your photogrammetric techniques. It would be possible to condense, but I would not say it is required. I will leave that up to you (the authors) to decide if you feel the level of detail is too much, or necessary to the overall description. The fact that there is significant landscape modification with extreme weather events is an interesting component - this locality survived Hurricane Joaquin, but it is certainly susceptible to future storms which could destroy this structure and thus its detailed documentation is warranted and appreciated.

We appreciate the editor considering the reviewers’ divergent recommendations (“accept as is” versus “major revisions”) and their advice in improving the manuscript, but also for siding with our fuller explanation of the scientific process underlying our study. As mentioned previously, a detailed report provides a framework for readers to better understand the discovery and subsequent diagnosis of this trace fossil. The photogrammetry is also meant to provide further evidence so our results can be examined, replicated, and tested by future generations. Moreover, considering the extensive damage wrought by Hurricane Jaoquin in October 2015 near the described outcrop, we thought our analysis was warranted as a precautionary measure. If this trace fossil is the only one of its kind on San Salvador Island and it is eroded by future events, the extra effort we took to document it will have been worth it.

Editor: You refer to 'root trace fossils' often, but this could be shortened to rhizoliths (sensu Kraus and Hasiotis, 2006: DOI: 10.2110/jsr.2006.052). It is a bit pedandic (no pun intended) but something to consider.

Although “rhizolith” is shorter than “root trace fossil,” it is not simpler. One of us (Martin) thoroughly considered the different terms used for trace fossils made by plant roots in Chapter 4 (Marginal Marine and Terrestrial Plants) in Life Traces of the Georgia Coast (2013, Indiana University Press), which minimally include rhizoliths, rhizomorphs, rhizocretions, and ichnorhizomorphs. In this review, he concluded that “root trace fossil” has the least ambiguity in meaning for geologists and paleontologists. Its use is thus similar to how the term “burrow” reflects an excavation made by an animal in sediment, regardless of its preservation, orientation, or other aspects. Accordingly, we prefer using “root trace fossils” throughout the text.

The resubmitted manuscript differs from the original in the following respects:

- Added subheadings to the Interpretation section and otherwise subtly altered the manuscript as suggested by both reviewers and the editor.

- Replaced Figure 1 so it consists solely of public-domain images and changed the figure caption to credit sources of these images.

- Included track changes in the revised document so all alterations are apparent to the editor.

- Included new links in Supplemental Files to animated videos of the photogrammetry model, which will make the model more accessible to readers with varying computer hardware and software.

---

## [Editor Report · Decision Letter 1]

12 Nov 2020

First Known Trace Fossil of a Nesting Iguana (Pleistocene), The Bahamas

PONE-D-20-22231R1

Dear Dr. Martin,

We’re pleased to inform you that your manuscript has been judged scientifically suitable for publication and will be formally accepted for publication once it meets all outstanding technical requirements.

Kind regards,

David M. Lovelace, Ph.D.

Academic Editor

PLOS ONE

Additional Editor Comments (optional):

Thank you for resubmitting your MS to PLOS ONE. It is my recommendation that this manuscript be accepted in its current form. Thank you for addressing the reviewers comments clearly and succinctly. Lastly, thank you for defending your use of 'root trace fossil' and the reference supporting your decision (I'll be adopting that in my own teachings/writings now).
---

## [Editor Report · Acceptance letter]

17 Nov 2020

PONE-D-20-22231R1 

First Known Trace Fossil of a Nesting Iguana (Pleistocene), The Bahamas 

Dear Dr. Martin:

I'm pleased to inform you that your manuscript has been deemed suitable for publication in PLOS ONE. Congratulations! Your manuscript is now with our production department. 

Kind regards, 

on behalf of

Dr. David M. Lovelace 

Academic Editor

PLOS ONE